# Refined cyclic renormalization group in Russian Doll model

Vedant Motamarri[1], Ivan M. Khaymovich[2,3], and Alexander S. Gorsky[4,5,6]

[1]TCM Group, Cavendish Laboratory, University of Cambridge, JJ Thomson Avenue, Cambridge, CB3 0HE, UK

[2]Nordita, Stockholm University & KTH Royal Institute of Technology, SE-106 91 Stockholm, Sweden

[3]Institute for Physics of Microstructures, Russian Academy of Sciences, 603950 Nizhny Novgorod, Russia

[4]Institute for Information Transmission Problems, Russian Academy of Sciences, 127051 Moscow, Russia

[5]Moscow Institute for Physics and Technology, Dolgoprudny 141700, Russia

[6]Laboratory of Complex Networks, Center for Neurophysics and Neuromorphyc Technologies, Moscow, Russia

We discuss the Russian Doll Model (RDM) of superconductivity for finite energy levels. Previously, cyclic renormalization group (RG) and Efimov scaling were found in RDM for part of the equidistant spectrum and we generalize this observation in a few directions. We find that when the whole spectrum is considered, equidistancy condition is removed or diagonal disorder is added, the cyclicity of RG survives but the period of RG becomes energy dependent. The analytic analysis is supported with exact diagonalization.

## Contents

Ivan M. Khaymovich: ivan.khaymovich@gmail.com, https://sites.google.com/view/ivan-khaymovich/

# 1 Introduction

The Richardson model of superconductivity [28, 29] is a suitable toy model with finite number of degrees of freedom which allows to capture the key properties of superconducting state in a relatively simple manner. The model is integrable and the corresponding spectrum can be obtained via the Bethe Ansatz (BA) equations which coincide with the BA equations for the twisted SU(2) Gaudin model [10]. The commuting Hamiltonians of the Richardson model get identified as superpositions of the Gaudin Hamiltonians.

There is a simple integrable deformation of the Richardson model – the so-called Russian Doll model (RDM) [19, 20] which involves the time-reversal (T) symmetry breaking deformation parameter. In this case, the BA equations for the spectrum are identical to the BA equations for the twisted inhomogeneous XXX $SU(2)$ spin chain. The inhomogeneities get identified with the energy levels of RDM model while the twist is the counterpart of the coupling constant in RDM. The T-symmetry breaking parameter in RDM is identified as the "Planck constant" in XXX spin chain which vanishes in the Gaudin limit [11]. The model can also be related to Chern-Simons theory when the excitations are represented by vertex operators [3].

The RDM model exhibits a rare property — it hosts cyclic renormalization group (RG) flows for the couplings [19]. The RG parameter is $\ln N$ where N is number of energy levels and the step in RG is "integrating out" the highest energy level. The coupling evolves in a cyclic manner while the T-symmetry breaking parameter fixes the period of the cycle. The RG cycle implies a nontrivial interplay between the ultraviolet- (UV) and infrared-limit (IR) physics, and the underlying algebraic property was identified as the anomalous breaking of scale invariance down to the discrete subgroup [2, 23].

Due to the remaining discrete scale invariance in the models with cyclic RG flows some part of the spectrum obeys the so-called Efimov exponential scaling $E_n \propto e^{cn}$. In the RDM, Efimov scaling has been found for the hierarchy of gaps, see [9] for a review. Note that recently new examples of cyclic RG [15], as well as examples of homoclinic RG orbits [14] and chaotic RG flows [7] have been found.

The ensemble averaging of the Richardson model with some measure for the diagonal on-site disorder has been considered in [18, 21, 25–27]. The measure for the ensemble averaging is a bit arbitrary and the relevance of the Generalized Gibbs Ensemble (GGE) for the disordered Richardson model has been discussed. The corresponding level statistics has been identified.

The effective Hamiltonian provides the possibility to elaborate the localization properties of the one-particle states. Of particular interest is whether our model shows the non-ergodic, but extended phase (usually called fractal or multifractal) for some values of $\gamma$ of the coupling strength $H_{m \neq n} \sim N^{-\gamma/2}$. Recall that fractal behavior has been observed in several models [4, 6, 8, 12, 17, 18, 22, 24–26, 30–32] and the Rosenzweig-Porter model (RPM) is the most familiar example, where multifractality has been found for the interval $1 < \gamma < 2$ [17]. Later the fractal behavior of RPM was confirmed within other approaches [6, 12, 22, 31, 32]. Using effective momentum-space RG and Fermi's Golden Rule, we have found that one-particle states in our version of RDM demonstrates a fractality interval as well [24]. The fractal dimension $D$ is determined exactly and equals $D = 1 - \gamma/2$.

In this study, we shall generalize the cyclic RG for couplings developed in [19] for the case of equidistant spectrum. First, we make a refinement of the cyclic RG for all energies for equidistant as well as non-equidistant spectra and find that the cyclic RG structure survives but the period of the RG becomes energy-dependent. The next aim is

the derivation of the RG for the random RDM with diagonal disorder. The analysis yields a similar result – the period of the RG becomes energy- and disorder-dependent. We also comment on the fate of the Efimov tower and the incomplete breaking of scale invariance in these cases.

Note that the T-breaking parameter is usually not renormalized perturbatively, but can be renormalized however if some kind of non-perturbative effects are taken into account. An example of RG in a disordered system with Anderson localization and T-breaking can be found in [1].

## 2 Model

We consider the $N_0 \times N_0$ random matrix model of the following form

$$H_{mn} = \varepsilon_n \delta_{mn} - j_{mn}, \quad j_{m \neq n} = \delta(N_0) \left[ g + ih \text{sign} \left( m - n \right) \right], \quad 1 \leq m, n \leq N_0 \ , \quad (1)$$

Note that we consider open boundary conditions and put the overall energy shift $j_{nn} = 0$ to zero without loss of generality. Here $\varepsilon_n$ is a certain (might be random and non-monotonic) potential of $n$ on a finite support

$$|\varepsilon_n| \leq \omega/2 \ . \quad (2)$$

## 3 Sierra's Renormalization group (RG) for all energies.

### 3.1 RG for equidistant spectrum

In Ref. [19] the authors consider the model (1) in the bosonic setting for application to superconductivity, with the following choice for the parameter

$$\delta(N_0) = \omega/N_0 \ , \quad (3)$$

with $\omega = \varepsilon_{N_0} - \varepsilon_1$ being the bandwidth [1] of the diagonal potential. They focused on the case with equidistant spectrum of the diagonal potential

$$\varepsilon_n = (n - n_0)\delta \quad (4)$$

with a certain energy shift $n_0 \delta$ (if not mentioned otherwise, we will use $n_0 = N_0/2$), giving the range of the diagonal energies as in (2). They derived the following renormalization group (RG), see Eq. (15) in Ref. [19], removing the largest diagonal energy level at each step:

$$g_{N-1} = g_N + \frac{g_N^2 + h_N^2}{N}, \ h_{N-1} = h_N \ . \quad (5)$$

In order to derive the above equations the authors of [19] did the following:

1. First, they start with the matrix of size $N_0$ and at each step reduce its size by one.

2. For this, they take at each step the level with the largest diagonal energy in the absolute value ($\varepsilon_N$ or $\varepsilon_1$).

---

[1] Unlike [19], we use $\omega$ for the total bandwidth, not its half and $\delta$ for the level spacing, not its half.

3. Assuming it to be large with respect to the rest of the levels, they resolve the eigen-problem with respect to it (say $\varepsilon_N$):

$$\left(\varepsilon_N - E\right)\psi_E(N) - \sum_n j_{Nn}\psi_E(n) = 0 \quad \Leftrightarrow \quad \psi_E(N) = \frac{\sum_{n \neq N} j_{Nn}\psi_E(n)}{\varepsilon_N - E} \tag{6a}$$

$$\left(\varepsilon_m - E\right)\psi_E(m) - \sum_n j_{mn}\psi_E(n) = 0 \quad \Leftrightarrow$$

$$\left(\varepsilon_m - E\right)\psi_E(m) - \sum_{n \neq N}\left(j_{mn} + \frac{j_{mN}j_{Nn}}{\varepsilon_N - E}\right)\psi_E(n) = 0 \tag{6b}$$

Strictly speaking, the latter fraction was split into two terms with $E$ replaced by $\varepsilon_m$ and $\varepsilon_n$, respectively, but this was not important for them.

4. Next, they assumed $\varepsilon_N - E \simeq \delta \cdot N$ and using the ratio $\omega/\delta = N$ they end up with Eqs. (5).

The solution of Eqs. (5) can be found in the continuous limit $ds \sim \Delta s = -\Delta N/N \ll 1$

$$h_N = h_{N_0} \equiv h \ , \tag{7a}$$

$$g_N = h \tan\left[hs_N + \arctan\left(\frac{g_{N_0}}{h}\right)\right] \ . \tag{7b}$$

with $\Delta N = 1$ and $s_N = \ln\left(N_0/N\right)$. Strictly speaking the above RG works for the bottom of the spectrum $E \sim \varepsilon_1$ if one takes the energies $\varepsilon_N$ always from the top of the spectrum.

## 3.2   Energy dependent RG periods

In general, one should replace the assumption in item 4 by the correct expression

$$g_{N-1} = g_N + \delta(N_0)\frac{g_N^2 + h_N^2}{\varepsilon_N - E}, \ \ h_N = h_0 \ . \tag{8}$$

Now the renormalization variable $s_E$ should be defined as

$$ds_E(N) = -\frac{\delta(N_0)}{\varepsilon_N - E} \quad \Leftrightarrow \quad s_E(N) = \sum_{n=N}^{N_0}\frac{\delta \cdot \Delta N}{\varepsilon_n - E} \approx \int_N^{N_0}\frac{\delta \cdot dn}{\varepsilon_n - E} \ , \tag{9}$$

where $\Delta N = 1$ and one arrives at the same RG equations and solution as Eqs.(15-16) in [19]

$$\frac{dg}{ds_E} = g^2 + h^2 \quad \Leftrightarrow \quad \boxed{g(s_E) = h \tan\left[hs_E + \arctan\left(\frac{g_{N_0}}{h}\right)\right]} \ . \tag{10}$$

The validity of the above equation (10) is limited by the conditions for the absence of resonance

$$|ds_E(N)| \ll 1, \frac{1}{g^2(s_E) + h^2} \quad \Leftrightarrow \quad |E - \varepsilon_N| \gg \delta, \delta \cdot \left[g^2(s_E) + h^2\right] \ . \tag{11}$$

The first condition $|ds_E(N)| \ll 1$ ensures that the increment in the integral (9) is small, while the second one limits the increment $|dg(s)|$ in (10) to make the derivation from (8) to it valid. Note that from Eq. (9) one can see that the monotonicity of the parameter $s$ (9) depends on the energy $E$ and does not necessarily require the monotonicity of $\varepsilon_n$. Indeed, for $|E| > \omega/2 > |\varepsilon_n|$, even random $\varepsilon_n$ does not change the monotonic behavior of $s_E(N)$. In the following two subsections, we apply the above considerations for equidistant and disordered diagonal potentials.

## 3.3 Entire spectrum for equidistant potential

For equidistant diagonal potential (4), one can introduce the following parameter $M_E = E/\delta + n_0$, for the energy shift, which gives:

$$s_E(N) = \ln\left(\frac{N_0 - M_E}{N - M_E}\right).$$

(12)

As in Eq.(17) of [19] the result (10) is periodic with the period $\lambda = \pi/h$ in $s_E$. The period $\lambda$ in $s_E$ corresponds to the change $\Delta N_T$ in the matrix size $N$ given by

$$\lambda = \ln\left(\frac{N - M_E}{N - \Delta N_T - M_E}\right) \;, \quad \boxed{\Delta N_T = (N - M_E)\left(1 - e^{-\lambda}\right).}$$

(13)

The number of periods before $N - M_E = 1$ goes as

$$\boxed{n = \frac{h}{\pi}\ln(N_0 - M_E).}$$

(14)

However, unlike [19], here we see two peculiarities:

- First, the period $\Delta N_T$ in $N$ is energy $E$-dependent, Eq. (13), and

- Second, there is the singularity at $N = M_E$, or equivalently, at $E = \varepsilon_N$.

The latter is important for the understanding of the following fact. Indeed, according to [19] and Eq. (13), the matrix size shrinks by $\Delta N_T(N)$ and the spectrum repeats itself with the shift by one level. However, on the way from $N_0$ to $N$ other $\Delta N_T - 1$ levels have also disappeared. Where have they gone?

To answer this physical question, one should consider the continuity condition (11) more closely. What happens when this condition is violated? In such a case, one cannot transform the sum in Eq. (9) to the integral and, moreover, already at one step one of the increments $ds_E(N)$ or $dg(s_E)$ is not small. This means that many periods can pass in this region without being seen in numerics. Strictly speaking, each RG step (8) corresponds to the removal of one column and one row of the matrix and can be considered as a rank-1 perturbation for the matrix [5]. As it is known from Richardson's model [21] and other works [16, 18], such a rank-1 perturbation can move significantly only one (top or bottom) level $E_N^{(N-1)}$, while the other $N - 2$ levels $E_n^{(N-1)}$ are bound in between the ones at the previous step

$$E_n^{(N)} < E_n^{(N-1)} < E_{n+1}^{(N)} \;.$$

(15)

This means that independently of the condition (11) only one level disappears from the spectrum by going to the drain at $E = \varepsilon_N$ in the RG step. We will show the same in our numerical results in Sec. 4.

## 3.4 Case of the diagonal disorder

Strictly speaking Eqs. (8) and (9) work for any diagonal potential, not only for equidistant or monotonic $\varepsilon_n$. Therefore in this subsection we consider disordered diagonal potential. In the case when the diagonal energies are not equidistant (4) but given by independent random numbers, the derivation of RG equations from (6) is not completely trivial.

In order to make the derivation clear let's consider separately the two effects of the disorder:

1. Fluctuations of $\varepsilon_n$ around their mean value (4);

2. Re-shuffling of $\varepsilon_n$.

Taking into account only the first effect, one can represent $\varepsilon_n$ as a sum of independent increments

$$\varepsilon_n = \varepsilon_1 + \sum_{k=1}^{n-1} \delta\varepsilon_k \ , \quad P(\{\delta\varepsilon_k\}) = \prod_{k=1}^{N-1} P_0(\delta\varepsilon_k) \ , \quad P_0(x) = \frac{1}{\delta} e^{-x/\delta} \ , \quad \langle \delta\varepsilon_k \rangle = \delta \ . \quad (16)$$

For large enough $n_E = n - n_0 - E/\delta \equiv n - M_E \gg 1$, Eq. (4), of i.i.d. random elements in the sum $\varepsilon_n - E$, can be approximated by a Gaussian random number with the following mean and variance

$$\langle \varepsilon_n - E \rangle = \delta \cdot n_E \ , \quad \sigma_{n,E}^2 = \left\langle (\varepsilon_n - E)^2 \right\rangle - \langle \varepsilon_n - E \rangle^2 = \delta^2 \cdot n_E \ , \quad (17)$$

and thus can be represented as

$$\varepsilon_n = E + \delta \cdot n_E + \delta \cdot \sqrt{n_E} g_n = \delta \left( n - n_0 \right) + \delta \cdot \sqrt{n_E} g_n \ , \quad (18)$$

with the standard Gaussian variable $g_n$

$$\langle g_n \rangle = 0 \ , \quad \left\langle g_n^2 \right\rangle = 1 \ . \quad (19)$$

The corresponding increment $ds_E(N)$, Eq. (9), is then given by

$$ds_E = \frac{\Delta N}{n_E \left( 1 + \sqrt{1/n_E} g_N \right)} \simeq \frac{\Delta N}{n_E} \left( 1 - \frac{g_N}{n_E^{1/2}} \right) \quad (20)$$

With the latter Taylor expansion this gives the result for $s_E(N)$ in terms of the central limit theorem as

$$s_E(N) = \ln \left( \frac{N_0 - M_E}{N - M_E} \right) - \tilde{g}_N \left( \sum_{n=N}^{N_0} \frac{1}{(n - M_E)^3} \right)^{1/2}$$

$$\simeq \ln \left( \frac{N_0 - M_E}{N - M_E} \right) - \frac{\tilde{g}_N}{\sqrt{2} \, |N - M_E|} \left[ 1 - \left( \frac{N - M_E}{N_0 - M_E} \right)^2 \right]^{1/2} \ , \quad (21)$$

Here we used the central limit theorem for the sum of Gaussians $g_n/n_E^{3/2}$ with zero means and variances $\sigma_n^2 = n_E^{-3}$ and introduced another standard Gaussian variable $\tilde{g}_N$, Eq. (19).

From the latter one can see that the additional summand $\sim |N - M_E|^{-1}$ to $s_N$ with respect to the one in the disorder-free case, Eq. (12), is small compared to the period $\pi/h$ for large enough $N - M_E$ within the RG validity region, Eq. (11). Strictly speaking, in the sum (21) one cannot keep the terms $O(1)$ as the Euler-Mascheroni constant $\gamma_E \simeq 0.5772$ is also neglected there.

At the same time, at the top of the spectrum (from where we take out $\varepsilon_N$) and close to $|N - M_E|$ the fluctuations will be important already at the level of Eq. (20). In the former region the central limit theorem in (18) does not hold, while in the latter the entire validity of the RG (11) is broken. As a result, the periodicity of the spectrum partly survives in the monotonic but disordered diagonal potential.

The reshuffling of the diagonal disorder has another effect. Indeed, as we mentioned in Sec. 3.2, in this case the monotonicity of the periodicity parameter $s_E(N)$ is guaranteed only for $|E| > \omega/2$. Therefore, the above periodicity (10) survives only in that region, while within the diagonal band, $|E| < \omega/2$, the parameter $s_E(N)$ can be non-monotonic with $N$ and random, and therefore no periodicity is expected.

## 3.5 Generalized Efimov scaling

Let us comment on the place of our study in the general context of the two-parametric RG flows when one parameter induces T-symmetry breaking. It is useful to introduce the following modular parameter [13]

$$\tau = x + iy, \text{ where } x \text{ is T-symmetry breaking term and } y \text{ is a some kind of disorder} \quad (22)$$

The real part is the chemical potential for the topological number of any nature, say, winding, topological charge etc. On the other hand the imaginary part is any parameter quantifying disorder, say, coupling constant, diffusion coefficient, boundary condition etc. In our case one could have in mind $\tau = h + ig$ while, for example , $\tau = \theta + iD$ for the Anderson model in 1d with T-symmetry breaking term $\theta$ and the diffusion coefficient $D$ [1]. Before renormalization there is the natural action of $SL(2, Z)$ on modular domain of $\tau$.

The pattern of RG orbits considered as trajectories of the dynamical systems depends on the relative weights of the perturbative and non-perturbative contributions to the $\beta$-functions. The conventional cyclic RG occurs at stable fixed point for $\text{Re}\,\tau$ taking into account only a first perturbative contribution to the $\beta_{\text{Im}\,\tau}$. The $\text{Re}\,\tau$ is finite at the stable fixed point and it governs the period of the RG cycle (10). Generically both the $\beta$'s are elliptic functions of the modular parameter and behave differently in the limiting cases.

The RG flow towards the stable fixed point can occur through the chain of unstable fixed points for $\text{Re}\,\tau$. For instance, such behavior and interesting universality has been observed in [13] in the limit $y = \text{Im}\,\tau \to 0$ when the potential function for the RG flow which yields the $\beta(x, y)$ function is the generalized Dedekind function.

$$U(x, y) = \log(y^{1/4}|\eta(x + iy)| \qquad \tau = x + iy , \quad (23)$$

where $\eta(z)$ is Dedekind function $\eta(z) = e^{\frac{\pi i z}{12}} \prod_{n=0}^{\infty} (1 - e^{2\pi i n z})$. At small fixed $y$ the RG potential for T-breaking parameter gets reduced to $U(x)$ whose minima $x_n$ exhibit the interesting recurrence $x_{n+1} = f(x_n)$ for the unstable critical points of the RG flow for $\text{Im}\,\tau$. The recurrence is ruled by the free group $\Gamma_2$ which is subgroup of $SL(2, Z)$ and involves three generators of the discrete RG flows [13]. At $n \to \infty$ the RG flows to the stable critical point while a topological parameter tends to the Golden ratio $x_n \to \frac{\sqrt{5}+1}{2}$. This model example corresponds to the one-dimensional Penrose model which is a toy model exhibiting cyclic RG cycle. In that case the Efimov scaling for the bound states reads as

$$E_n = E_0 \exp(cn) \quad (24)$$

with $c = \ln(\frac{\sqrt{5}+1}{2})$. In the refined RG, once again we look at the stable fixed point of $\text{Re}\,\tau$ but the period of $\text{Im}\,\tau$ is energy dependent

$$g(s_E + \lambda) = g(s_E) . \quad (25)$$

A bit loosely we could say that the $\text{Re}\,\tau$ defining the period at the fixed point is E-dependent. Instead of (24) we have scaling of the form

$$\frac{\log(\frac{E_n}{E_0})}{s(E_n)} \sim n \quad (26)$$

which reduces to Efimov scaling for equidistant spectrum. Note that the Efimov scaling follows from the partial breaking of the scale invariance down to the discrete subgroup [2, 23]. In the refined case the discrete subgroup is broken as well.

## 4    Numerics

In order to check the analytical predictions of the previous section, we have performed numerical simulations similar to [19]. Taking the initial Hamiltonian (1) of size $N_0$, we compute the spectrum for the models, given by the first $N$ rows and $N$ columns of the matrix. Then the spectrum of such models (normalized by the parameter $\delta$) has been plotted versus the periodicity parameter $s_E(N)$, Eq. (9), see Figs. 1 – 3. For our numerics we have chosen $N_0 = 256$, $g = 1$, and $h = 12$, though the results are qualitatively the same for other parameters as well.

In the case of the equidistant spectrum, Fig. 1, the periodicity parameter is given by (12), $s_E(N) = \ln\left(\frac{N_0 - M_E}{N - M_E}\right)$, with $M_E = E/\delta + n_0$. One can see from Fig. 1 that the periodicity differs in different regions of the spectrum, but it is still given by the above formula. At the spectral edges (see the first and last panels), the energy levels may disappear with decreasing $N$ (and increasing $s_E(N)$). In addition, close to the energies $E/\delta = N \equiv \varepsilon_N/\delta$, (see the bottom part of the last panel in Fig. 1), the periodicity is violated in full agreement with the validity range, Eq. (11). Note that it is not just the discreteness of the spectrum which matters as for smaller energies $|E|/\delta < N$ even the discrete spectrum shows the same periodicity, cf. left ($|E|/\delta < N$) and right ($|E|/\delta > N$) bottom panels.

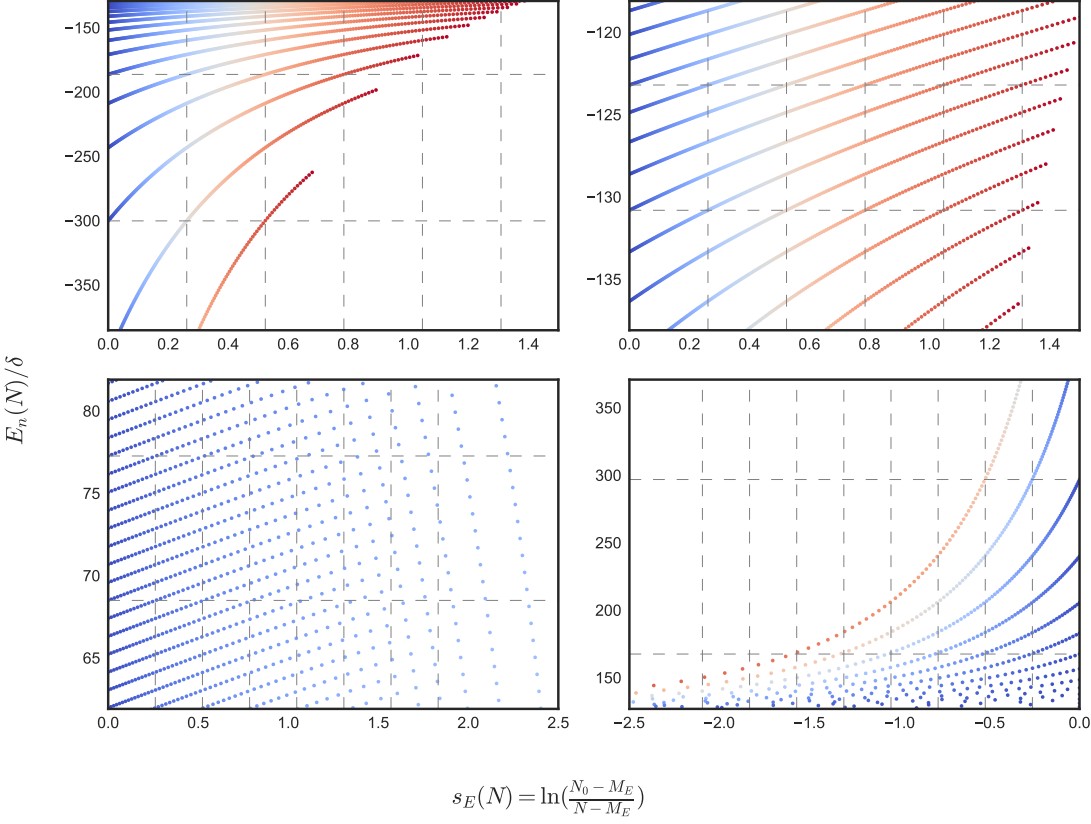

$$s_E(N) = \ln\left(\frac{N_0 - M_E}{N - M_E}\right)$$

Figure 1: **Generalized $E$-dependent spectrum periodicity, Eq. (12), in the Russian Doll model with equidistant diagonal potential, Eq.** (4) **in different spectral parts.** Vertical lines correspond to the periodicity in the parameter $s_E(N) = \ln\left(\frac{N_0 - M_E}{N - M_E}\right)$, which perfectly matches the one in the numerical spectrum in all those parts.

In the more physical and interesting case of disordered diagonal potential (17), one

has to modify the periodicity parameter to (9) or in the monotonic case to (21). In this case, see Fig. 2, the periodicity is still clearly seen, but close to the interval of the diagonal potential energies, $|E| < \omega/2$ (with $n_0 = N_0/2$) the periodic levels are not seen under the ones with random shifts along $s_E(N)$. The latter are those levels, which hit the resonance $E \simeq \varepsilon_N$ and, thus, have non-monotonic $s_E(N)$ vs $N$.

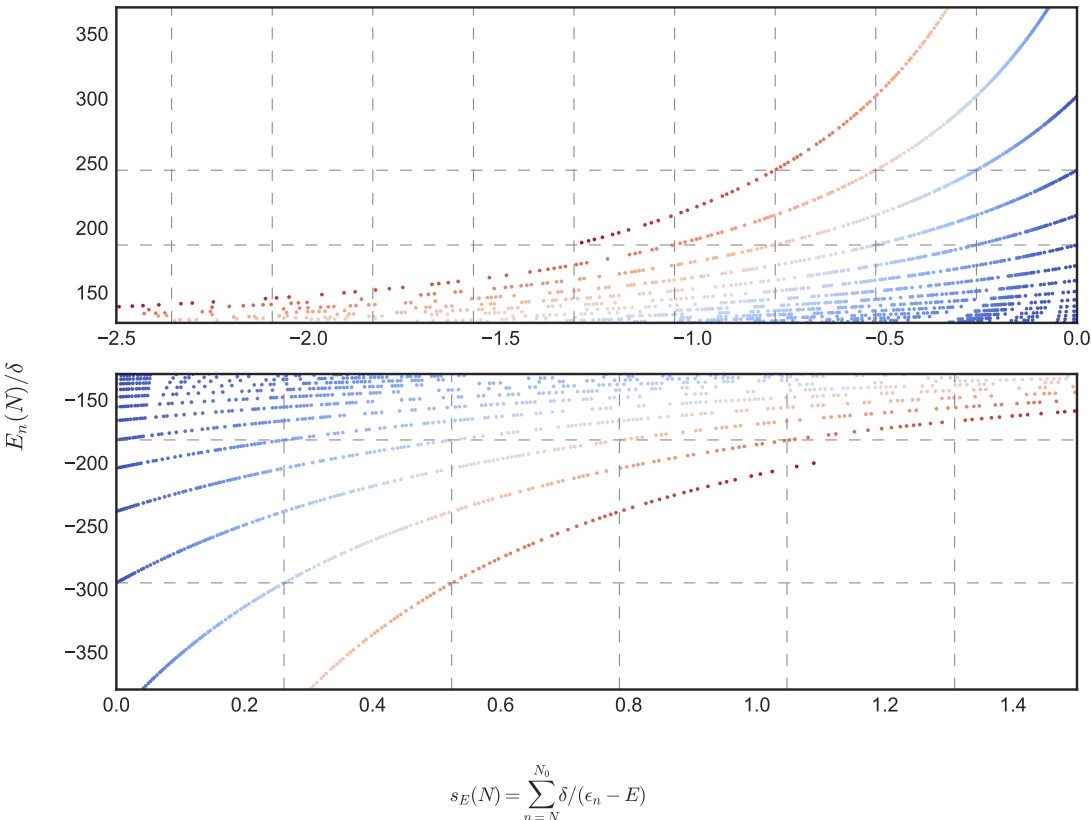

$$s_E(N) = \sum_{n=N}^{N_0} \delta/(\epsilon_n - E)$$

Figure 2: **Generalized $E$-dependent spectrum periodicity, Eq.** (17)**, in the Russian Doll model with random diagonal potential** in different spectral parts. Vertical lines show the periodicity in the parameter $s_E(N)$ (9), which provides reasonable match to the periodicity of the spectrum in the parts, away from the diagonal potential bulk, $|E| > \omega/2$.

In order to show clearly the range of random energies, we plot the entire spectrum of the system in Fig. 3 versus the local periodicity parameter $s_E(N) = \sum_{n=N}^{N_0} \delta/(\varepsilon_n - E)$. From that figure the periodicity is hard to see due to the symbol sizes, but one can clearly observe that in the interval $|E|/\delta < N_0/2$ the random levels, corresponding to the above hitting of resonances and non-monotonic $s_E(N)$, prevails over the the regular ones, so the latter are not seen. Beyond the above mentioned energy interval, i.e. for $|E|/\delta > N_0/2$, no such random levels are visible.

In addition, a random singular point of the spectrum appears at $E = \varepsilon_N$, where the regular spectral part changes behavior from $s_N > 0$ at $E < \varepsilon_N$ to $s_N < 0$ otherwise. These are exactly the sink points which are random within the interval $|\varepsilon_N| < \omega/2$ at each step $N$ where most of the levels disappear beyond RG periodicity.

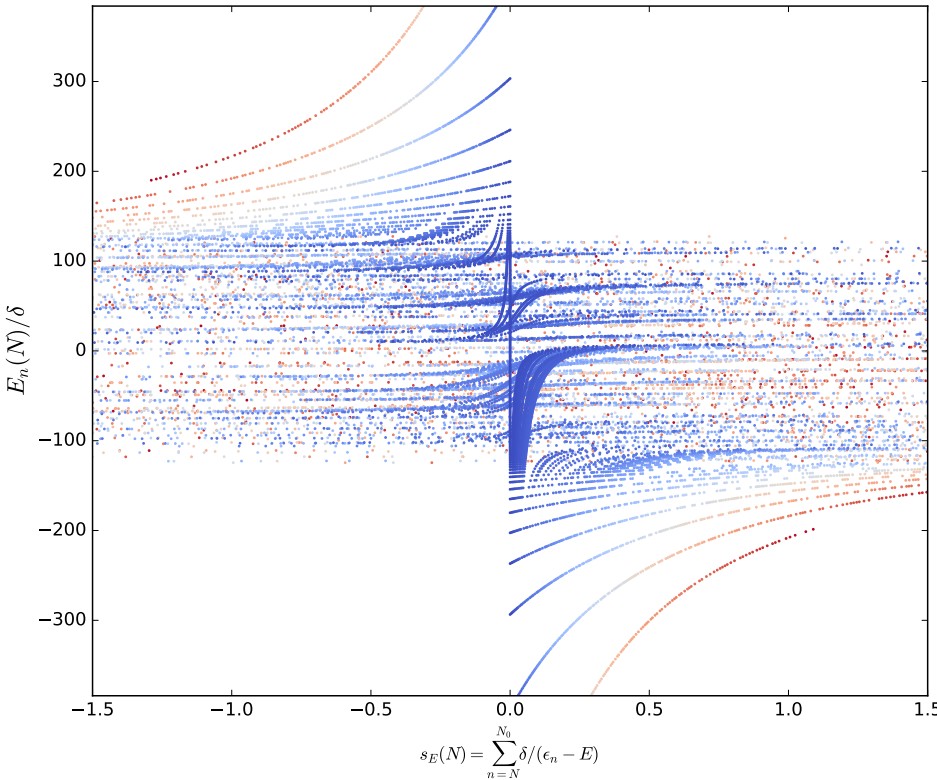

Figure 3: **Overview of generalized $E$-dependent spectrum periodicity in the Russian Doll model with random diagonal potential.** One can clearly see that the periodicity is broken within the diagonal potential bulk, $|E| < \omega/2$.

## 5   Conclusion

In this paper we have generalized the periodic renormalization group (RG) for the known Russian Doll model (RDM) in several ways.

In the original RDM with the equidistant diagonal elements, we have shown that the RG period depends significantly on the energy interval and has a singularity at the sink point $E = \varepsilon_N$. It is this singularity which compensates the disbalance of $\Delta N_T - 1$ energy levels that should disappear after the RG period $\Delta N_T \simeq$ and, according to the previous literature [19, 20], shift the entire spectrum by one level only.

In addition, we have considered the RDM with the disordered diagonal elements and found a generalized RG parameter over which RG equations are still periodic, at least in the spectral parts lying beyond the energy interval of the diagonal elements. We have considered both the effects of small oscillations of the diagonal elements as well as their re-shuffling due to disorder.

All the analytical predictions have been confirmed by the numerical simulations. It would be interesting to identify the limit cycle breaking discussed in this study with the generic framework of breakdown of the limit cycle in the bifurcation theory.

# 6  Acknowledgments

V. M. thanks MPIPKS for providing an opportunity to take part in the Summer Internship program during which this project was initiated and acknowledges support from the Harding Foundation. I. M. K. acknowledges support by the Russian Science Foundation (Grant No. 21-72-10161). A. S. G. thanks Nordita and IHES where the parts of this work have been done for the hospitality and support.

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
