# Peer review of "Refined cyclic renormalization group in Russian Doll model"

_SciPost Physics_

## Round 1 · Referee Report · Anonymous (Referee 1) · 2024-10-15

Strengths

  1. This is an interesting extension of the Russian Doll BCS model.

  2. It points out how naive Efimov scaling is not necessarily exact but can depend on the energy.

  3. Extensive numerical support.

Weaknesses

  1. See report for minor revisions.

Report

I recommend this paper for publication after a few minor modifications.

The authors study an interesting extension, or further study of the Russian Doll BCS model. They point out that Efimov exponential scaling can be violated. Namely the RG period can be energy dependent.

Requested changes

  1. The QFT model in Ref. [20] has the same RG beta function and Russian Doll resonances, where their masses are $M_n = 2m \cosh( n \lambda )$ where $\lambda$ is the RG period. This only has exact Efimov scaling for large n: $M_n = m e^{n \lambda}$ Thus at low energies we don't have exact Russian Doll scaling.

Can the authors comment on whether this is related to their main result?

  1. Section 3 is named "Sierra's RG..." and refers to Refs. [19][20]. However he is one of 2 other authors in these papers, so referring to it as Sierra's RG should be corrected.

Recommendation

Ask for minor revision

---

## Round 1 · Referee Report · Anonymous (Referee 2) · 2024-10-16

Strengths

  1. Interesting subject potentially useful for many problems
  2. Simple enough analysis (compared to Bethe ansatz)
  3. Validity of results

Weaknesses

  1. Introduction is not guided by physics and difficult to digest even by prepared readers

Report

I recommend to publish to paper after a major revision of the presentation, especially Introduction

Requested changes

Rewrite Introduction in more physics terms

Attachment

Recommendation

Ask for major revision

---

## Editorial Decision

resubmitted